# Verify What Matters: Budgeted Verification for Tool-Using Agents under Counterfactual Downstream Harm

## Abstract

Tool-using agents make intermediate decisions that alter persistent state, shape later observations, and create failures that are not equally easy to recover from. When verification is costly, the central question is not whether checking helps in general, but which decisions are worth checking. Policies driven only by local uncertainty capture whether a step may be wrong, but not how much that error would matter if left uncorrected. We formulate *budgeted verification* for tool-using agents as an intervention-allocation problem in which the value of checking a step depends on verifier efficacy, local error probability, downstream harm, and intervention cost. This factorization makes uncertainty-only routing a restricted approximation appropriate when downstream harm is nearly constant, but incomplete when actions differ in persistence, reversibility, and dependency structure. Empirically, we evaluate this framing in a controlled OpenClaw-facing sandbox, not a full gateway-paired OpenClaw benchmark. On the dependency-sensitive slice with an exact budget of 150 verifications over 300 episodes, uncertainty-only routing achieves 0.813 success, 0.187 irreversible-failure rate, and 3.455 average net reward, while harm-aware routing under the same verify rate achieves 0.817 success, 0.183 irreversible-failure rate, and 3.700 average net reward. The paired bootstrap intervals for these main differences overlap zero, so we do not present this table as a large-margin benchmark result. Instead, the evidence is diagnostic: harm-aware routing selects higher-consequence episodes, $H$-only is the strongest ablation, and $p \times H$ improves over $p$-only while not outperforming the structural harm cue alone. Additional diagnostics include structural-proxy alignment with exact branch harm, a multi-budget curve at 10%, 25%, 50%, and 75% verification rates, pairwise ablation confidence intervals, cross-slice behavior, verifier-quality sensitivity, a disjoint held-out seed split, a small true held-out procedural variant, a monotone-binned calibration probe, learned PRM-style baselines, and two stress tests showing failure modes under harm-cue corruption and correlated verifier failures. Taken together, the results support the paper's central allocation claim in a bounded form: under scarce verification, local uncertainty alone is an incomplete routing signal, and downstream consequence must be modeled explicitly. The current transparent $p \times H$ score should be viewed as an interpretable diagnostic instantiation rather than a universally dominant router.

## 1 Introduction

Language-model agents are increasingly deployed in environments where they must act rather than only generate text. They browse websites, call tools, edit files, and execute multi-step procedures whose outcome depends on a sequence of intermediate decisions (Yao et al., 2023; Schick et al., 2023; Liu et al., 2023; Zhou et al., 2024; Xie et al., 2024; Jimenez et al., 2024; Kinniment et al., 2023). In these settings, reliability is not only a question of whether the final answer is correct: intermediate actions can change the environment state, shape later observations, and create failures that are easy or hard to recover from (Ruan et al., 2024; Andriushchenko et al., 2025; Rabanser et al., 2026). This turns verification into a resource-allocation problem. When checking a step is costly, the system cannot verify everything, so it must decide which decisions are worth checking.

A natural response is to use uncertainty: if a step looks uncertain, ask for confirmation, invoke a verifier, or pause before committing (Kadavath et al., 2022; Lin et al., 2022; Ren et al., 2023; Farquhar et al., 2024). This is sensible, and uncertainty is an important baseline. But in a tool-using trajectory, uncertainty alone does not determine the value of intervention. Two steps may have similar local error likelihood while differing sharply in downstream consequence. A reversible formatting choice and a state-changing tool action need not be treated the same way, even if both look equally uncertain at the moment they are proposed. The distinction matters because tool-using trajectories are stateful. An incorrect intermediate action can do more than reduce the probability of final success: it can corrupt working state, alter what later steps observe, or trigger execution failures whose downstream effects may be difficult to repair (Grinsztajn et al., 2021; Peng et al., 2026). The missing quantity is therefore not another confidence score, but counterfactual downstream harm.

This paper studies *budgeted verification* from that perspective. We treat verification as a scarce intervention that should be allocated according to expected downstream loss reduction. The core idea is simple: the value of checking a step depends on the probability that the step is wrong, the downstream harm that error would cause if left uncorrected, the probability that verification successfully repairs the error, and the cost of intervening. The factorization $q_t p_t H_t - c_t$ makes verification a structured decision problem in which local error likelihood, downstream consequence, verification efficacy, and intervention cost play different roles. Uncertainty-only routing corresponds to the special case in which $H_t$ is treated as constant; the framework is broader precisely when that assumption fails.

This framing is broader than any single agent runtime, but it is especially natural for tool-using agents with persistent state and real execution side effects. We use OpenClaw (OpenClaw, 2026) as a motivating and evaluation substrate, while keeping the paper's claim narrower than a full benchmark study. Our experiments isolate the ranking principle in a controlled one-opportunity-per-episode setting where the exposed public uncertainty signal does not always align with downstream consequence. The central empirical question is focused: when only some decisions can be checked, does routing verification by downstream harm protect the right episodes better than routing by uncertainty alone?

**Contributions.** We make four contributions. **(i)** We formulate *budgeted verification* for tool-using agents as an intervention-allocation problem over state-changing trajectories, separating a step's local error likelihood from the downstream harm of leaving that error uncorrected. **(ii)** We introduce a consequence-aware verification value, $V_t = q_t p_t H_t - c_t$, that clarifies when uncertainty-only routing is a special case and when it is not, and we identify when PRM-thresholding is a legitimate instance of the framework and when it is not. **(iii)** We give a concrete *deployment recipe* (§3.6) covering estimator choice, calibration data, harm-weight selection, and exact-budget threshold tuning. **(iv)** We report a controlled OpenClaw-facing evaluation suite at $n$=300 for the main sandbox cells, including an exact-budget dependency-sensitive comparison, structural harm-proxy alignment with exact branch harm, a multi-budget curve, component ablations with paired intervals, cross-slice routing behavior, verifier-quality sensitivity, learned PRM-style baselines, a disjoint seed-split diagnostic, a small true held-out procedural diagnostic, and stress tests for harm-cue corruption and correlated verifier failures.

**Scope of evidence.** The empirical claim is deliberately bounded, but not merely negative. The results are controlled OpenClaw-facing sandbox experiments, not a full gateway-paired OpenClaw benchmark. They test allocation under fixed episode generation, fixed policy access, and one eligible verification point per episode. Within that setting, the exact-budget comparisons hold verification volume fixed across policies. The completed evidence does not show that the uncalibrated product $\hat{p}_t \hat{H}_t$ is always the best finite-sample router. It does show that downstream consequence is a necessary routing signal that uncertainty-only policies miss, and it identifies the conditions under which a transparent consequence-aware rule succeeds or fails. The structural harm cue is a proxy for the theoretical $H_t$, not a calibrated counterfactual estimator; the new proxy-alignment diagnostic below evaluates this relationship rather than assuming it. The one-step routing rule is not an optimal solution to the fully online multi-step budgeted control problem. These limits are part of the paper's contribution: they separate the decision-theoretic target from the empirical proxy and make the assumptions testable. We revisit them in §6.

## 2 Related Work

**Tool-using agents, uncertainty, and process supervision.** Recent work has expanded language models into interactive systems that reason, act, and call external tools (Yao et al., 2023; Schick et al., 2023), with benchmarks such as AgentBench, WebArena, OSWorld, and SWE-bench (Liu et al., 2023; Zhou et al., 2024; Xie et al., 2024; Jimenez et al., 2024; Kinniment et al., 2023) providing multi-step evaluation environments. We use OpenClaw as an execution substrate (OpenClaw, 2026). Orthogonal lines study *uncertainty and calibration* in language models that express confidence (Guo et al., 2017; Kadavath et al., 2022; Lin et al., 2022; Farquhar et al., 2024), conformal-prediction frameworks that decide when to request human help (Ren et al., 2023), and uncertainty propagation, clarification, and tool-use miscalibration in agents (Wang et al., 2025b; Zhao et al., 2025; Wang et al., 2025a; Xuan et al., 2026). *Process supervision* treats intermediate steps as first-class evaluation objects (Lightman et al., 2023; Wang et al., 2023; Setlur et al., 2025), with extensions to process reward models (PRMs) for sequential decision making (Choudhury, 2025; Xi et al., 2025; Wang et al., 2025a) and step-level tool-use benchmarks (Li et al., 2026; Fan et al., 2026). These papers address *whether* a step is likely wrong; under a limited budget, however, the most uncertain step is not always the most valuable to verify.

**When PRM-thresholding is a special case, and when it is not.** Under the composite loss $\mathcal{L}_{\text{task}} + \beta\mathcal{L}_{\text{harm}}$, the limit $\beta=0$ reduces the decision objective to task-success ranking, exactly what a PRM trained on task success approximates; thresholding a task-success PRM is then a legitimate instance of our framework. A natural follow-up is whether a *harm-weighted* PRM, trained to predict a composite score reflecting both task success and irreversible-side-effect cost, would dominate the explicit $\hat{p}_t \cdot \hat{H}_t$ decomposition for $\beta>0$. We take this question seriously: the present paper compares against a harm-weighted PRM baseline trained on the same features as our structural harm cue in §4.9. The conceptual boundary is that a task-trained PRM cannot distinguish task failure from irreversible side effects when those carry different weights, but a harm-weighted PRM can absorb the composite signal into a single scalar at the cost of separately interpretable inputs; we report the empirical comparison rather than claiming either dominates in principle.

**Self-correction, safety, monitoring, and selective intervention.** Reflexion, Self-Refine, and CRITIC (Shinn et al., 2023; Madaan et al., 2023; Gou et al., 2024) show that corrective feedback helps once applied but do not resolve allocation. A growing literature studies harm at the level of agent behavior via sandbox evaluations (Ruan et al., 2024), harm benchmarks (Andriushchenko et al., 2025; Zhang et al., 2025), runtime safeguards (Xiang et al., 2025; Chen et al., 2025; Hua et al., 2024), contextual-harm monitoring (Inan et al., 2025; Korbak et al., 2025; Luo et al., 2025), tool-execution hallucination analysis (Peng et al., 2026), and injection-defense patterns (Beurer-Kellner et al., 2025; Rabanser et al., 2026); these establish downstream harm as a deployment concern but do not formalize what happens when the verification budget is tight. The closest conceptual neighbor is selective prediction and cost-aware decision making (Chow, 1970; Geifman & El-Yaniv, 2017; 2019; Xin et al., 2021; Janisch et al., 2019; Zellinger & Thomson, 2025), with deeper roots in the cost-sensitive learning literature (Elkan, 2001; Domingos, 1999; Bartlett & Wegkamp, 2008), where Elkan's foundational treatment shows that optimal cost-sensitive decisions decompose into per-class probability times per-class cost, the same factorization our framework specialises to the step-level intervention setting. Connections to safe RL (Amodei et al., 2016; García & Fernández, 2015; Grinsztajn et al., 2021) and scalable oversight (Bai et al., 2022; Burns et al., 2023) share our perspective that intervention is worthwhile only when benefit justifies cost, but the setting differs: tool-using agents operate in sequential, state-changing environments where an early mistake propagates, and the decision is not whether to abstain on a single prediction but whether to verify an intermediate step whose effect depends on trajectory-level propagation. We view the present paper as applying step-level cost-sensitive decision theory to budgeted runtime verification, with novelty in the operational $p \cdot H$ separation, the matched-budget evaluation design, and the deployment recipe of §3.6, rather than in the underlying expected-utility decomposition.

**Relation to recent agentic reasoning and allocation systems.** Recent agentic reasoning systems and benchmarks emphasize complementary parts of the tool-use reliability problem. AlphaApollo studies multi-turn agentic reasoning and self-evolution with tool-assisted verification as part of a propose–judge–update loop (Zhou et al., 2026a); AR-Bench evaluates whether models can actively acquire missing information

under incomplete evidence (Zhou et al., 2025a); and Landscape of Thoughts visualizes reasoning trajectories and can adapt trajectory features into a lightweight verifier (Zhou et al., 2025b). CoDaPO is closer in spirit to resource allocation, but it allocates training updates by confidence and difficulty under a fixed compute budget rather than allocating runtime verification to proposed state-changing actions (Zhou et al., 2026b). Our contribution is orthogonal: given proposed tool actions and a fixed oversight budget, we separate local error likelihood from downstream consequence and ask which actions should receive verification.

Our paper is not another uncertainty estimator, self-correction module, agent benchmark, or runtime safeguard. It is a budgeted intervention problem: given limited opportunities to check actions, which decisions should receive verification?

## 3 Method

### 3.1 Problem Setup

We consider a tool-using agent that acts over a finite horizon of $T$ steps. At step $t$, the agent has access to a trajectory history $h_t = (o_1, \bar{a}_1, o_2, \bar{a}_2, \ldots, o_t)$, where $o_t$ is the current observation and $\bar{a}_{1:t-1}$ are the previously executed actions. Given this history, the base agent proposes a step action $a_t \sim \pi_A(\cdot \mid h_t)$, which may denote a tool call, a structured intermediate action, or any other committed decision that can affect future reasoning or the external environment. Before that action is executed, a verification policy decides whether to check it, $v_t \in \{0, 1\}$. If verification is skipped, the executed action is $\bar{a}_t = a_t$; if verification is applied, the system may confirm or correct the proposal, yielding $\bar{a}_t$, and the environment transitions using the executed action rather than the raw proposal. Verification is therefore an *intervention* on the trajectory rather than an abstention: when the system skips verification, it commits the proposed step and continues.

Let $\tau$ denote the resulting trajectory. The episode loss is

$$\mathcal{L}(\tau) = \mathcal{L}_{\text{task}}(\tau) + \beta \, \mathcal{L}_{\text{harm}}(\tau),$$

where $\mathcal{L}_{\text{task}}$ captures task failure or degraded quality and $\mathcal{L}_{\text{harm}}$ captures downstream side effects such as corrupted state, irreversible mistakes, or repair burden, with $\beta \geq 0$ controlling how strongly harmful effects are weighted. Each verification incurs cost $c_t \geq 0$ (latency, extra tool calls, human oversight), and we study the constrained problem

$$\min_{\pi_V} \, \mathbb{E}\big[\mathcal{L}(\tau)\big] \qquad \text{subject to} \qquad \mathbb{E}\Big[\sum_{t=1}^{T} c_t v_t\Big] \leq B.$$

### 3.2 Local Error Probability and Counterfactual Downstream Harm

The central modeling choice is to separate *how likely the current step is to be wrong* from *how much damage that error would cause if left uncorrected*. Let $E_t \in \{0, 1\}$ indicate whether the proposed step $a_t$ is erroneous relative to the task and environment semantics at $h_t$, and define the local error probability $p_t = \Pr(E_t = 1 \mid h_t, a_t)$. This is a step-local quantity: it measures how plausible it is that the current proposal is wrong, but not whether that error is cheap to recover from or costly to allow.

To capture consequence, suppose that $E_t = 1$. Consider two continuations sharing the same prefix up to step $t-1$: an *error continuation* in which the incorrect proposal is executed and the trajectory proceeds from the resulting state, and a *corrected continuation* in which step $t$ is replaced by an appropriate corrected action $a_t^{\text{corr}}$. Let the resulting future trajectories be $\tau_{t:T}^{\text{err}}$ and $\tau_{t:T}^{\text{corr}}$. We define the downstream harm magnitude at step $t$ as

$$H_t = \mathbb{E}\big[\mathcal{L}(\tau_{t:T}^{\text{err}}) - \mathcal{L}(\tau_{t:T}^{\text{corr}}) \mid h_t, a_t, E_t = 1\big].$$

$H_t$ is the expected future loss caused by leaving the current error uncorrected, defined *conditional on the step actually being wrong*. It is therefore conceptually distinct from $p_t$. A step can have high local error probability but low downstream harm if its effects are easy to detect and repair later; conversely, a step can have moderate local error probability but very large downstream harm if it writes persistent state, alters what later steps observe, or creates a failure that subsequent reasoning will trust and amplify. Uncertainty-only

verification focuses on $p_t$ alone, which is reasonable when harm is roughly uniform across steps; in the settings that motivate this paper, however, that approximation collapses step-to-step differences in consequence that determine whether spending verification is worthwhile.

**From theoretical $H_t$ to an operational proxy.** We flag the gap between the definition above and the empirical instantiation. $H_t$ as defined is a counterfactual expectation over future trajectories under error and correction, which in general requires either a learned dynamics model or Monte-Carlo rollouts to estimate. The experiments in this paper do not estimate $H_t$ in this sense: they use a hand-engineered *structural harm cue* (§3.5) computable from the proposed tool call's declared metadata. This cue is best described as a *correlate* of $H_t$ on the task distribution we evaluate on. A concrete failure mode: two `file.write` operations with identical declared class and rollback flag can have very different true $H_t$ if one writes a temporary artifact that downstream steps discard while the other writes a canonical artifact that downstream reasoning trusts. The structural cue cannot distinguish these cases; $H_t$, in principle, can. Readers should treat the theoretical framework (this section) and the empirical instantiation (§4) as addressing related but non-identical quantities, and we return to this in §6 and Appendix D.

## 3.3 Verification Value

Let $q_t \in [0, 1]$ denote verifier efficacy at step $t$, defined as the probability that verification successfully catches and corrects the step when $E_t=1$. The expected reduction in future trajectory loss obtained by verifying step $t$ is then $\Delta_t = q_t p_t H_t$. Verification matters only if the current step is wrong (probability $p_t$), only if the verifier catches and repairs that error (probability $q_t$), and the amount of avoided future loss conditional on repair is the downstream harm term $H_t$. After subtracting intervention cost, the net verification value is

$$V_t = q_t \, p_t \, H_t - c_t.$$

This is the ideal decision target. It clarifies what uncertainty alone can and cannot tell us: even a well-calibrated uncertainty signal does not distinguish a low-impact error from a high-impact one unless consequence is modeled separately. If $H_t$ were constant across steps and $q_t$ were fixed, the rule would collapse to uncertainty ranking; the additional modeling content lies in allowing harm magnitude to vary across steps. When verification costs differ, the ranking uses the value-per-cost score $S_t = q_t p_t H_t / (c_t + \epsilon)$; in the equal-cost setting studied empirically here, this reduces to ranking by $q_t p_t H_t$.

**Definition 1 (Myopic step-level decision target).** We define the step-level myopic decision target as $V_t^{\text{myopic}} = q_t p_t H_t - c_t$, with the corresponding rule $v_t^{\text{myopic}} = \mathbf{1}[q_t p_t H_t \geq c_t]$, or under a Lagrangian relaxation with multiplier $\lambda > 0$, $v_t^\star = \mathbf{1}[q_t p_t H_t \geq \lambda c_t]$. This is a definition, not a derivation of the optimal online policy. Three assumptions make this target locally correct: (i) verification affects the trajectory only by replacing the current proposed action with a corrected one when verification succeeds, (ii) verification cost is additive, and (iii) *future verification choices are held fixed*. Assumption (iii) is restrictive: it is exactly the assumption that suppresses sequential coupling between verification decisions across steps, and as a consequence the myopic rule above is *not* in general the optimal online policy under a hard budget constraint. The full multi-step problem cannot be solved by this ranking rule alone; a proper sequential treatment requires either dynamic programming over a tractable state reduction or a regret-style analysis, which we do not attempt here (Appendix A).

What the definition does contribute is fixing a per-step decision target whose factor structure, error probability, harm magnitude, verifier efficacy, and intervention cost, is interpretable and estimable in deployment. As soon as $H_t$ varies across steps, pure uncertainty ranking is suboptimal even for this myopic target. We frame the empirical results as testing routing behaviour relative to this target, not as testing optimality against a sequential oracle.

## 3.4 Online Budgeted Verification Policy

In deployment, $p_t$, $H_t$, and $q_t$ are not directly observed. The verification policy operates on estimates $\hat{p}_t, \hat{H}_t, \hat{q}_t$ derived from features $x_t = \phi(h_t, a_t, b_t)$ that include uncertainty signals, action type, state-mutation indicators,

dependency information, and rollback availability. The resulting online score is $\hat{V}_t = \hat{q}_t \hat{p}_t \hat{H}_t - \lambda(b_t, t)\, c_t$, and the policy verifies iff $\hat{V}_t \geq 0$, where $\lambda(b_t, t)$ is a budget-control term that tightens as budget is depleted. What should remain invariant across implementations is the decision *structure*: verification should be triggered by an estimate of intervention value, not by uncertainty alone. The decomposition uses a product of estimates, so miscalibration in any component can affect the final ranking; a slight overestimate of $\hat{p}_t$ and a slight overestimate of $\hat{H}_t$ compound, which is why the deployment recipe below calibrates the components *jointly* on held-out data. A learned composite $\hat{V}_t$ can reduce this compounding at the cost of separately-interpretable inputs (Appendix B).

**Choosing the harm weight $\beta$.** The composite loss $\mathcal{L}_{\text{task}} + \beta \mathcal{L}_{\text{harm}}$ requires a choice of $\beta$ encoding how aggressively the deployment prefers to avoid irreversible side effects relative to ordinary task failure. This is not solved by the method; it is a deployment decision. As a heuristic, if a stakeholder can accept one irreversible failure per $K$ task failures, then a natural starting choice is $\beta$ of order $K$. A worked example clarifies the scale: consider a deployment in which an unrecovered task failure costs an operator \$0.50 in retry resources and customer-relations overhead, while an irreversible side effect (say, an erroneous database write that requires manual reconciliation) costs \$200 in downstream engineering time. The stakeholder ratio is then $K \approx 400$, suggesting an initial $\beta$ near 400 in cost-unit-comparable terms; a deployment with weaker irreversibility cost (say a content-moderation agent where an erroneous flag can be reviewed in \$2 of human time) would warrant $K \approx 4$, an order-of-magnitude difference. In the $\beta \to 0$ limit, the framework reduces to task-reward ranking; in the $\beta \to \infty$ limit, it verifies every step with any nonzero harm cue, which is budget-infeasible at scale. Intermediate values are the regime where consequence-aware allocation is meaningful, and Appendix B discusses how this choice interacts with PRM-thresholding. In practice, deployments should report sensitivity over a plausible range of $\beta$ values rather than claiming $\beta$-invariant dominance; the experiments below should therefore be read as one operational reward setting, not as a universal loss model.

## 3.5 Operational Estimation in the Sandbox

The experiments instantiate the ideal quantities deliberately simply. The goal is not to claim that counterfactual harm estimation is solved, but to make the allocation rule executable and inspectable. The local error proxy $\hat{p}_t$ is the public uncertainty signal exposed at the eligible decision; it is available to every policy and captures the ambiguity visible to the agent before committing. In other deployments, the same role could be played by self-consistency, a log-probability margin, a step-error classifier, or a calibrated verifier-disagreement signal; we use the public signal because it gives uncertainty-only routing the cleanest possible baseline. The harm proxy $\hat{H}_t$ is a structural cue computed from the proposed tool action and the current trajectory prefix using declared side-effect class, downstream dependency count, and rollback availability; the exact functional form is given in Table 1. It does not read the slice label, the ground-truth outcome, or the realized future trajectory. For analysis only, the sandbox also records an exact branch-harm value after the episode is generated; this is never used by the online policy and is reported only to check whether a routing rule selected higher-consequence episodes on average. Verifier efficacy $q_t$ is treated as a controlled detection probability, with default $q{=}0.35$ and sensitivity checks at $q \in \{0.20, 0.35, 0.50, 0.80\}$. For exact-budget comparisons, thresholds are chosen so that the competing policies spend the same number of verifications; on the main dependency-sensitive slice, both `uncertainty_only_matched_budget` and `harm_aware_selective` verify exactly 150 of 300 episodes.

This is a restricted one-opportunity-per-episode setting, but it is sufficient to test the central allocation question under a clean budget constraint. The next subsection turns the framework into a concrete recipe.

## 3.6 Deployment Recipe

A persistent reviewer concern with consequence-aware verification frameworks is that they specify a decision target without telling practitioners how to estimate the components or what to optimise on validation data. We address this directly. Algorithm 1 is a six-step recipe for taking the framework from a new setting to a deployable threshold. Each step has a concrete output, a calibration objective, and a checkable invariant.

Table 1: Implementation specification of the operational proxies used in the sandbox. The structural harm cue and public uncertainty signal use only quantities computable from the trajectory prefix and the proposed action's declared metadata.

| Quantity | Implementation |
|---|---|
| $\hat{p}_t$ (public uncertainty) | Scalar in $[0, 1]$ returned by the agent's exposed confidence head at the eligible decision; specifically, $1 - \max_k \pi_A(a_t^{(k)} \mid h_t)$ over the top-3 candidate proposals, clipped to $[0, 1]$. |
| $\hat{H}_t$ (structural harm cue) | $\hat{H}_t = w_{\text{cls}} \cdot \text{class}(a_t) + w_{\text{dep}} \cdot \text{dep}(a_t, h_t) + w_{\text{rb}} \cdot \mathbf{1}[\neg \text{rollback}(a_t)]$, with $w_{\text{cls}}{=}0.4$, $w_{\text{dep}}{=}0.4$, $w_{\text{rb}}{=}0.2$. $\text{class}(a_t) \in \{0, 0.5, 1\}$ for {read-only, write-recoverable, write-irreversible}; $\text{dep}(a_t, h_t) \in [0, 1]$ is the normalised count of declared downstream consumers; rollback is a declared boolean. |
| $\hat{q}_t$ (verifier efficacy) | Bernoulli with controlled rate $q \in \{0.20, 0.35, 0.50, 0.80\}$; verifier detection events are independent across episodes. |
| $c_t$ (verification cost) | Fixed at 0.5 reward units per verification. |
| Exact branch harm (analysis only) | The realised difference $\mathcal{L}(\tau_{t:T}^{\text{err}}) - \mathcal{L}(\tau_{t:T}^{\text{corr}})$ computed by the sandbox after episode generation. Never read by any online policy; used only to verify that a routing rule selected higher-consequence episodes on average. |

---

**Algorithm 1** Deployment recipe for budgeted verification.

---

1: **Step 1: Define harm.** Specify $\mathcal{L}_{\text{harm}}$ as a function of trajectories on a representative episode set. For tool-using agents, a useful default is an indicator of irreversible state change weighted by repair cost. Output: a deterministic per-episode harm score.

2: **Step 2: Choose $\beta$.** Solicit a stakeholder ratio $K =$ "tolerable task failures per tolerable irreversible failure"; set $\beta \approx K$. Sanity-check by inspecting the implied loss on a held-out batch and, where possible, repeat validation over a plausible range of $\beta$ values.

3: **Step 3: Select estimators.** Pick a $\hat{p}_t$ source (public confidence, log-prob margin, self-consistency, or a learned step-error classifier) and a $\hat{H}_t$ source (a structural cue as in Table 1, a learned predictor trained on counterfactual rollouts, or a hybrid). Estimate $\hat{q}_t$ from offline verifier evaluation, preferably conditionally on action type, side-effect class, rollback availability, harm class, state, and verifier/model family rather than as a single global accuracy.

4: **Step 4: Joint calibration on a held-out set.** On a held-out split $\mathcal{D}_{\text{cal}}$ of $\geq 200$ episodes, fit per-component monotone calibrators (isotonic regression) for $\hat{p}_t$ against realised $E_t$ and for $\hat{H}_t$ against realised exact branch harm. Calibrate *jointly*: hold $\hat{q}_t$ fixed and choose isotonic maps for $\hat{p}_t, \hat{H}_t$ that minimise the deviation of $\hat{q}_t \hat{p}_t \hat{H}_t$ from realised $\Delta_t$ in Brier score. Joint calibration matters because per-component miscalibration compounds multiplicatively.

5: **Step 5: Fix the budget.** Set a verification budget $B$ as a verify-rate cap (e.g. "verify at most 50% of state-changing actions") or an expected-cost cap. The cap encodes the available oversight capacity and is independent of the routing rule.

6: **Step 6: Threshold on the validation objective.** On $\mathcal{D}_{\text{cal}}$, sweep thresholds $\theta$ for $\hat{V}_t$ and choose $\theta^\star$ minimising $\widehat{\mathcal{L}}_{\text{task}} + \beta \widehat{\mathcal{L}}_{\text{harm}}$ subject to the budget cap. Deploy with $v_t = \mathbf{1}[\hat{V}_t \geq \theta^\star]$.

---

This recipe is the deployment procedure recommended by the framework. Our main experiments instantiate its budget-fixing and threshold-selection principles, and §4.8 provides a seed-disjoint calibration diagnostic.

Two invariants should hold across the recipe. First, joint calibration is non-negotiable: per-component calibration is not sufficient because the routing score is a product and a calibrator fit on each marginal does not control the joint. Second, the budget is fixed *before* the threshold is tuned. This is what isolates allocation quality from verification volume; a policy that improves outcomes only by verifying more often is not exhibiting the property we care about. The exact-budget experiments in §4 instantiate Step 5 by enforcing a verify-rate of 0.500 and instantiate Step 6 by choosing the threshold that achieves that rate.

The same principle applies to verifier reliability. A single average $\hat{q}_t$ can be misleading when verifier failures concentrate on high-harm actions. The correlated-failure diagnostic in §4.11 shows this explicitly: a verifier can have a higher aggregate detection rate while rarely correcting the hard high-harm cases. We therefore recommend estimating or auditing $\hat{q}_t$ conditionally on action type, side-effect class, rollback availability, harm class, state, and verifier/model family, with separate reporting on high-harm cases.

## 4 Experiments

### 4.1 Experimental Setup

Our empirical goal is to evaluate allocation under scarce verification. The question is not whether verification helps when applied everywhere, but whether a limited verification budget is spent on episodes where an unchecked error would cause larger downstream loss. We therefore separate two protocols. In exact-budget comparisons, policies verify the same number of episodes and differ only in which episodes they select. In shared-threshold cross-slice diagnostics, thresholds are held fixed so that verification rate can vary with the slice's risk structure.

We use a controlled OpenClaw-facing sandbox. Each episode contains one eligible verification point before a final state-changing action. At that point, a policy can inspect the public uncertainty signal and, depending on the policy, structural information about the proposed tool action. Verification incurs a fixed cost of 0.5 reward units. If the verifier detects the hidden issue, the proposed action is replaced by a corrected action specified by the episode. The agent model, episode generator, reward decomposition, and verifier interface are held fixed across policies. The quantitative tables come from this controlled sandbox runner. A representative embedded OpenClaw trace validates interface compatibility, but it is not used as a quantitative benchmark result.

We evaluate three main implemented slices and one small held-out procedural variant. `easy` contains low-harm recoverable episodes. `dependency_sensitive` is the main diagnostic slice, where public uncertainty is not always aligned with downstream consequence. `corruption_prone` contains globally high-harm actions. After the main results were locked, we added a minimal `dep_sensitive_held_out` case with shifted structural-harm cues and report it only as a small diagnostic stress test. Separately, where calibration or learned baselines are needed, we use a disjoint held-out seed split within `dependency_sensitive`: seeds 20–119 for calibration or fitting and seeds 120–319 for evaluation.

For paired comparisons, we use paired percentile bootstrap intervals over the seed index with 10,000 resamples. Positive differences in success and reward favor the first policy; negative differences in irreversible failure favor the first policy. Unless stated otherwise, bootstrap intervals for the main positive comparisons overlap zero, so we use the language of directional or diagnostic evidence rather than statistical dominance. In the tables, boldface marks the directionally preferred value within the relevant matched comparison block. Boldface is descriptive and is not a significance marker.

**Reproducibility artifacts.** An anonymized supplementary repository is available at `https://anonymous.4open.science/r/bva-26-agent-supplement-4DE2`. It contains the controlled OpenClaw-facing sandbox runner, policy runlists, raw per-episode outputs, aggregation scripts, bootstrap scripts, calibration and PRM scripts, locked summary files, checksum records, example configuration files, and a dedicated OpenClaw interface folder. The interface folder contains the reset/hint and finalize scripts, episode templates, prompts, runlists, and saved embedded trace outputs used to validate the agent-facing workflow. The controlled evaluation preserves the agent-level structure relevant to this paper: persistent workspace state, proposed tool actions, hidden execution issues, verifier interventions, corrected actions, downstream rollouts, and reward consequences. This design lets us evaluate routing policies under paired seeds and matched verification budgets. A larger gateway-paired OpenClaw benchmark over unconstrained natural tasks is an important external-validity extension, but the quantitative claims in this revision are tied to the controlled OpenClaw-facing evaluation.

Table 2: Exact-budget controlled evaluation on `dependency_sensitive` ($n$=300). Both selective policies verify exactly 150 episodes. Branch harm is recorded after episode generation for analysis only and is not read by the online routing policy.

| Policy | Verify rate | Success | Irrev. fail | Avg. net reward | Sel. harm | Skip. harm |
|---|---|---|---|---|---|---|
| `uncertainty_only_matched_budget` | 0.500 | 0.813 | 0.187 | 3.455 | 11.100 | 11.490 |
| `harm_aware_selective` | 0.500 | **0.817** | **0.183** | **3.700** | **11.590** | **10.999** |
| $\Delta$ (harm-aware − uncertainty) | 0.000 | +0.003 | -0.003 | +0.245 | +0.490 | -0.491 |
| Paired bootstrap 95% CI | n/a | [-0.027, 0.033] | [-0.033, 0.027] | [-0.625, 1.163] | n/a | n/a |

Table 3: Multi-budget diagnostic on `dependency_sensitive`. Budgets correspond to exact top-$B$ verification among 300 episodes.

| $B$ | Verify rate | Policy | Success | Irrev. fail | Avg. net reward | Selected harm |
|---|---|---|---|---|---|---|
| 30 | 0.10 | $p$ only | 0.680 | 0.320 | -0.842 | 12.528 |
| 30 | 0.10 | $H$ only | 0.683 | 0.317 | -0.611 | 14.832 |
| 30 | 0.10 | $p \times H$ | **0.687** | **0.313** | **-0.582** | **15.124** |
| 75 | 0.25 | $p$ only | 0.723 | 0.277 | 0.602 | 10.785 |
| 75 | 0.25 | $H$ only | 0.727 | 0.273 | 0.856 | 11.803 |
| 75 | 0.25 | $p \times H$ | **0.740** | **0.260** | **1.194** | **13.154** |
| 150 | 0.50 | $p$ only | 0.813 | 0.187 | 3.455 | 11.100 |
| 150 | 0.50 | $H$ only | **0.827** | **0.173** | **4.065** | **12.320** |
| 150 | 0.50 | $p \times H$ | 0.817 | 0.183 | 3.700 | 11.590 |
| 225 | 0.75 | $p$ only | 0.900 | 0.100 | 6.164 | 11.011 |
| 225 | 0.75 | $H$ only | **0.920** | **0.080** | **6.928** | **12.031** |
| 225 | 0.75 | $p \times H$ | 0.907 | 0.093 | 6.483 | 11.436 |

## 4.2 Exact-Budget Dependency-Sensitive Evaluation

Table 2 reports the primary exact-budget comparison on `dependency_sensitive`. Both policies verify exactly 150 of 300 episodes. Under this fixed budget, harm-aware routing slightly improves success, irreversible failure, and average net reward relative to uncertainty-only routing. More importantly for the allocation claim, it selects episodes with higher average exact branch harm and leaves lower average branch harm unverified. The outcome margins are small, and the paired bootstrap intervals for success, irreversible failure, and net reward include zero. We therefore do not treat this table as a large-margin performance claim. We use it for the claim it can support: when verification volume is fixed, adding downstream consequence changes which episodes are protected and moves the harm-sensitive outcomes in the predicted direction.

## 4.3 Multi-Budget Dependency-Sensitive Diagnostic

To check that the allocation behavior is not an artifact of the single 50% operating point in Table 2, Table 3 evaluates deterministic exact top-$B$ routing on the same `dependency_sensitive` slice at budgets of 10%, 25%, 50%, and 75%. The table compares $p$-only, $H$-only, and $p \times H$ rankings using the same 300 episodes and reconstructs the verified and unverified continuations from the locked per-episode counterfactual outputs. Verified episodes use the corrected continuation; unverified episodes use the wrong continuation. This table is a diagnostic of budget-dependent allocation, not a new benchmark claim.

Across all four budgets, the consequence-aware rankings select higher exact branch harm than $p$-only routing. The product $p \times H$ is directionally strongest at the tighter 10% and 25% budgets, where combining local error likelihood with consequence is most useful under scarce oversight. At the larger 50% and 75% budgets, $H$-only is directionally strongest, reinforcing the ablation finding that downstream consequence is the dominant

Table 4: Cross-slice behavior under shared routing thresholds ($n$=300 per case-policy pair). Verification rates are intentionally not forced to match across slices.

| Case | Policy | Verify rate | Verify n | Success | Irrev. fail | Avg. net reward | Sel./skip harm |
|------|--------|-------------|----------|---------|-------------|-----------------|----------------|
| easy | harm_aware_selective | **0.000** | **0** | 1.000 | 0.000 | 9.200 | 0.000 / 0.000 |
| easy | uncertainty_only_matched_budget | 0.247 | 74 | 1.000 | 0.000 | 9.200 | 0.000 / 0.000 |
| easy | random_verify_matched_budget | 0.523 | 157 | 1.000 | 0.000 | 9.200 | 0.000 / 0.000 |
| dependency_sensitive | harm_aware_selective | 0.500 | 150 | **0.817** | **0.183** | **3.700** | **11.590 / 10.999** |
| dependency_sensitive | uncertainty_only_matched_budget | 0.503 | 151 | 0.813 | 0.187 | 3.455 | 11.026 / 11.567 |
| dependency_sensitive | random_verify_matched_budget | 0.523 | 157 | 0.807 | 0.193 | 3.204 | 10.124 / 12.580 |
| corruption_prone | harm_aware_selective | 1.000 | 300 | **1.000** | **0.000** | **9.200** | **31.160 / 0.000** |
| corruption_prone | uncertainty_only_matched_budget | 0.870 | 261 | 0.890 | 0.110 | 5.020 | 31.011 / 32.154 |
| corruption_prone | random_verify_matched_budget | 0.523 | 157 | 0.597 | 0.403 | -6.127 | 30.255 / 32.154 |

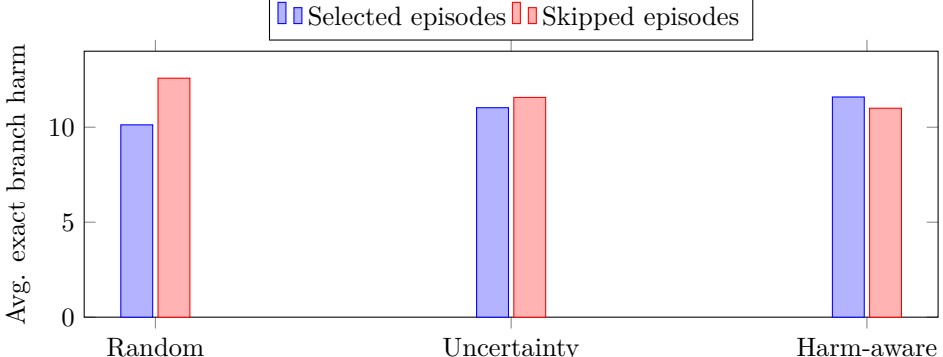

Figure 1: Selected-versus-skipped branch harm on `dependency_sensitive`. Branch harm is analysis-only. The figure illustrates allocation behavior rather than statistical significance.

finite-sample routing signal in this sandbox. This pattern supports consequence-aware allocation while also limiting the empirical claim about any particular uncalibrated product score.

### 4.4 Cross-Slice Behavior under Shared Thresholds

Table 4 evaluates whether the same routing thresholds change verification rate in the expected direction across slices. This is not an exact-budget table. On `easy`, harm-aware routing verifies no episodes and still obtains perfect success. On `dependency_sensitive`, harm-aware routing is slightly better than uncertainty-only at approximately matched budget. On `corruption_prone`, harm-aware routing escalates to verifying all episodes because the slice is uniformly high harm. We use this table as a routing-behavior diagnostic rather than as the primary fairness comparison.

Figure 1 visualizes the dependency-sensitive allocation pattern. Harm-aware routing pulls selected harm upward and skipped harm downward relative to uncertainty-only, although the separation is modest at $n$=300.

### 4.5 Structural Harm-Proxy Alignment

The theoretical quantity $H_t$ is a counterfactual downstream-harm expectation, while the implemented router uses the structural proxy $\hat{H}_t$ in Table 1. To make this gap explicit, Table 5 reports Pearson and Spearman alignment between the structural harm cue and exact branch harm recorded after episode generation. Correlations are computed on unique episodes; selected-versus-skipped harm is computed for the harm-aware router.

The pooled alignment is strong because the structural cue separates low-harm, dependency-sensitive, and corruption-prone regimes. Within `dependency_sensitive`, however, the rank association is weak ($\rho$=0.149), although the harm-aware router still selects episodes with higher exact branch harm on average (11.590 vs. 10.999). We therefore interpret $\hat{H}_t$ as a coarse structural routing cue rather than a calibrated estimator of $H_t$.

Table 5: Alignment between the structural harm proxy and exact branch harm. Pearson and Spearman correlations are computed on unique episodes. Selected/skipped harm is computed for the harm-aware router. "n/a" indicates a constant slice or an empty selected/skipped group.

| Slice | $n$ | Pearson $r$ | Spearman $\rho$ | Selected harm | Skipped harm |
|---|---|---|---|---|---|
| all slices | 900 | 0.688 | 0.733 | 24.637 | 3.666 |
| corruption_prone | 300 | n/a | n/a | 31.160 | n/a |
| dependency_sensitive | 300 | 0.093 | 0.149 | 11.590 | 10.999 |
| easy | 300 | n/a | n/a | n/a | 0.000 |

Table 6: Component ablation on dependency_sensitive under the same exact budget ($n$=300, 150 verifications per policy).

| Policy | Verify rate | Verify n | Success | Irrev. fail | Avg. net reward | Sel./skip harm |
|---|---|---|---|---|---|---|
| $p$ only | 0.500 | 150 | 0.813 | 0.187 | 3.455 | 11.100 / 11.490 |
| $H$ only | 0.500 | 150 | **0.827** | **0.173** | **4.065** | **12.320 / 10.269** |
| $p \times H$ | 0.500 | 150 | 0.817 | 0.183 | 3.700 | 11.590 / 10.999 |

This is the intended scope of the empirical instantiation: it tests whether consequence information changes allocation under a fixed budget, not whether general counterfactual harm estimation is solved.

## 4.6 Component Ablation of Routing Signals

Table 6 asks whether the result is driven by uncertainty, downstream harm, or their product. All three policies verify exactly 150 of 300 dependency-sensitive episodes. The ablation is important because it narrows the paper's empirical claim. $H$-only is directionally strongest, $p \times H$ improves over $p$-only, and the uncalibrated product does not outperform $H$-only. Table 7 adds paired bootstrap intervals for the component differences. These intervals cross zero, so we do not claim statistical dominance. The correct interpretation is that downstream consequence is the dominant signal in this sandbox, while the available public uncertainty proxy adds limited marginal ranking value.

## 4.7 Imperfect-Verifier Sensitivity

The sandbox treats verifier detection as a controlled probability and evaluates $q \in \{0.20, 0.35, 0.50, 0.80\}$. Table 8 shows that both policies improve as verifier quality increases. Harm-aware routing has directionally higher average net reward at all tested $q$ values under the same exact budget. However, at the weakest verifier setting, success and irreversible-failure rates are essentially tied and slightly favor uncertainty-only. Bootstrap intervals for the paired differences cross zero, so this table should be read as sensitivity evidence rather than universal dominance.

## 4.8 Disjoint Held-Out Seed Split and Joint Calibration

We first use a disjoint seed split within dependency_sensitive to test calibration and learned routing models. This seed-disjoint diagnostic is different from the small true dep_sensitive_held_out procedural variant reported below. Seeds 20–119 form the calibration or training split; seeds 120–319 form the evaluation split. The exact budget is 100 verifications over 200 held-out evaluation episodes.

Table 9 reports the calibration diagnostic. A pure-Python monotone-binning calibrator is fit on the calibration split, avoiding any additional package dependency. The calibrated $p \times H$ score changes the selected top-$B$ set relative to the uncalibrated score, with Jaccard overlap 0.626, but it does not improve aggregate evaluation outcomes. The important conclusion is diagnostic: calibration changes ranking decisions, yet in this split it does not close a measurable outcome gap. The product remains directionally better than $p$-only and $H$-only on this held-out split, but bootstrap intervals include zero.

Table 7: Paired bootstrap confidence intervals for component ablations at exact budget $B{=}150/300$ on `dependency_sensitive`. Deltas are first policy minus second policy. Positive success, negative irreversible failure, higher reward, and higher selected harm favor the first policy.

| Comparison | Metric | $\Delta$ | 95% CI |
|---|---|---|---|
| $H$ only $- p$ only | Success | +0.013 | [-0.030, 0.057] |
| $H$ only $- p$ only | Irrev. fail | -0.013 | [-0.057, 0.030] |
| $H$ only $- p$ only | Avg. net reward | +0.610 | [-0.687, 1.957] |
| $H$ only $- p$ only | Selected harm | +1.220 | [-0.862, 3.384] |
| $H$ only $- p \times H$ | Success | +0.010 | [-0.023, 0.043] |
| $H$ only $- p \times H$ | Irrev. fail | -0.010 | [-0.043, 0.023] |
| $H$ only $- p \times H$ | Avg. net reward | +0.365 | [-0.624, 1.391] |
| $H$ only $- p \times H$ | Selected harm | +0.730 | [-0.770, 2.282] |
| $p \times H - p$ only | Success | +0.003 | [-0.027, 0.033] |
| $p \times H - p$ only | Irrev. fail | -0.003 | [-0.033, 0.027] |
| $p \times H - p$ only | Avg. net reward | +0.245 | [-0.650, 1.155] |
| $p \times H - p$ only | Selected harm | +0.490 | [-1.026, 2.025] |

Table 8: Verifier-quality sensitivity on `dependency_sensitive` ($n{=}300$ per row). Both policies verify exactly 150 episodes for every $q$ value.

| $q$ | Policy | Verify rate | Success | Irrev. fail | Avg. net reward |
|---|---|---|---|---|---|
| 0.20 | `uncertainty_only_matched_budget` | 0.500 | **0.797** | **0.203** | 3.007 |
| 0.20 | `harm_aware_selective` | 0.500 | 0.790 | 0.210 | **3.047** |
| 0.35 | `uncertainty_only_matched_budget` | 0.500 | 0.813 | 0.187 | 3.455 |
| 0.35 | `harm_aware_selective` | 0.500 | **0.817** | **0.183** | **3.700** |
| 0.50 | `uncertainty_only_matched_budget` | 0.500 | 0.847 | 0.153 | 4.302 |
| 0.50 | `harm_aware_selective` | 0.500 | **0.860** | **0.140** | **4.774** |
| 0.80 | `uncertainty_only_matched_budget` | 0.500 | 0.953 | 0.047 | 7.093 |
| 0.80 | `harm_aware_selective` | 0.500 | **0.957** | **0.043** | **7.209** |

## 4.9 Comparison with Learned PRM-Style Routers

A natural objection is that the framework could be replaced by a learned scalar router. We test this with two pure-Python logistic baselines trained on the calibration split. `PRM-task` predicts unverified task failure. `PRM-value` predicts a normalized positive reward gain from verification. Both use tabular features derived from the same uncertainty and harm components used by the transparent routers. Table 10 shows that the learned routers are directionally stronger than fixed $p \times H$ on the held-out split, although paired bootstrap intervals cross zero. This result does not undermine the consequence-aware framing. It suggests that learned allocation models can absorb the same consequence signal when counterfactual calibration labels are available, while the transparent product remains a non-learned and interpretable baseline.

## 4.10 Small True Held-Out Procedural Diagnostic

In addition to the seed-disjoint diagnostic above, we implemented a small held-out procedural variant, `dep_sensitive_held_out`, after the main `dependency_sensitive` results had been locked. The patch adds a new sandbox case with shifted structural-harm cue distribution and leaves the locked `dependency_sensitive` experiments unchanged. We treat this result as a stress test of cross-generator robustness rather than as a headline positive result.

Table 11 reports an exact top-$B{=}50/100$ matched-budget comparison on seeds 400–499. The result is mixed: $p$-only is directionally strongest, while $H$-only and $p \times H$ remain close. All paired bootstrap intervals cross zero. This diagnostic therefore does not show that $p \times H$ generalizes best. Instead, it shows that cross-generator

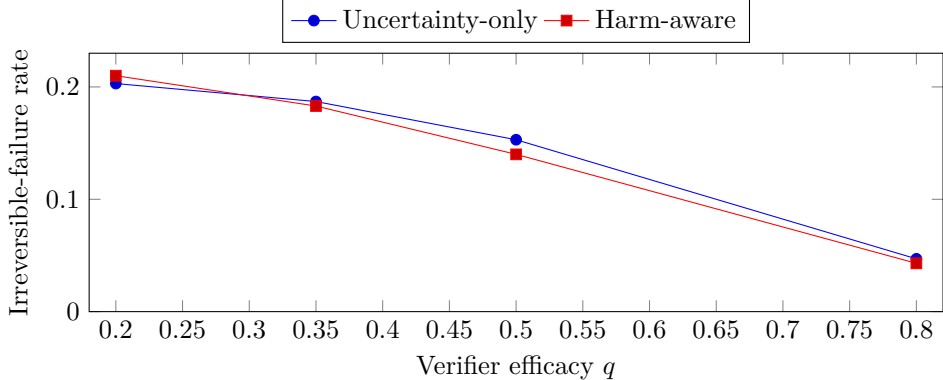

Figure 2: Irreversible-failure rate as a function of verifier efficacy $q$ under matched exact budget. Differences are small, especially at the weakest and strongest verifier settings.

Table 9: Joint-calibration diagnostic on a disjoint held-out seed split within `dependency_sensitive`. Evaluation uses seeds 120–319 ($n$=200) and an exact budget of 100 verifications.

| Policy | Verify rate | Success | Irrev. fail | Avg. net reward |
|---|---|---|---|---|
| $p$ only | 0.500 | 0.790 | 0.210 | 2.871 |
| $H$ only | 0.500 | 0.790 | 0.210 | 3.092 |
| $p \times H$ | 0.500 | **0.795** | **0.205** | **3.172** |
| calibrated $p \times H$ | 0.500 | **0.795** | **0.205** | **3.172** |

performance depends on whether the held-out generator preserves the alignment among public risk, structural harm cues, and downstream consequence.

### 4.11 Stress Tests: Harm-Cue Corruption and Correlated Verifier Failures

The preceding results support consequence-aware routing as an allocation principle, but they also reveal two concrete failure modes. First, consequence-aware routing depends on reliable harm estimates. Table 12 adversarially relabels the highest-harm evaluation episodes by setting their harm score to the minimum evaluation harm score before selecting the exact top 100 episodes. Performance degrades sharply as the corrupted fraction increases. Unlike the main positive comparisons, the bootstrap intervals for reward degradation and irreversible-failure increase exclude zero. This is best read as a limitation: systematic corruption of the harm cue can make consequence-aware routing fail.

Second, average verifier quality is not sufficient to characterize robustness. Table 13 compares a uniform $q$=0.35 verifier with a correlated verifier whose failures concentrate on high-harm selected cases. The correlated verifier has a higher overall detected rate in this diagnostic, but it rarely corrects hard high-harm cases, and aggregate outcomes degrade. Bootstrap intervals exclude zero for the degradation in success, irreversible failure, and reward. This indicates that verifier reliability should be modeled jointly with downstream consequence rather than treated as a single global constant.

### 4.12 Tight-Budget Corruption-Prone Diagnostic

Finally, we run an exact-budget diagnostic on `corruption_prone`. Under a tight budget of 150 verifications over 300 episodes, harm-aware and uncertainty-only select exactly the same top-$B$ seed set, with Jaccard overlap 1.0, and therefore obtain identical outcomes: 0.590 success, 0.410 irreversible failure, and $-6.380$ average net reward. This is not positive evidence. It is a boundary case showing that when harm is nearly uniformly high, the structural harm cue provides little ranking separation beyond the uncertainty signal.

Table 10: Learned PRM-style routing baselines on the same held-out seed split as Table 9. The learned baselines are directional diagnostics, not statistically conclusive dominance claims.

| Policy | Verify rate | Success | Irrev. fail | Avg. net reward |
|---|---|---|---|---|
| $p$ only | 0.500 | 0.790 | 0.210 | 2.871 |
| $H$ only | 0.500 | 0.790 | 0.210 | 3.092 |
| $p \times H$ | 0.500 | 0.795 | 0.205 | 3.172 |
| PRM-task | 0.500 | **0.805** | **0.195** | 3.256 |
| PRM-value | 0.500 | **0.805** | **0.195** | **3.324** |

Table 11: Small true held-out procedural diagnostic on `dep_sensitive_held_out`. Evaluation uses seeds 400–499 ($n$=100) and an exact budget of 50 verifications. This is a diagnostic stress test, not conclusive cross-generator dominance evidence.

| Policy | Verify rate | Success | Irrev. fail | Avg. net reward | Selected harm | Skipped harm |
|---|---|---|---|---|---|---|
| $p$ only | 0.500 | **0.860** | **0.140** | **4.318** | **13.473** | **9.764** |
| $H$ only | 0.500 | 0.840 | 0.160 | 4.024 | 12.885 | 10.352 |
| $p \times H$ | 0.500 | 0.840 | 0.160 | 3.922 | 12.681 | 10.556 |

## 5 Discussion

The experiments support the central allocation argument in a precise form. Under an exact verification budget, harm-aware routing is directionally better than uncertainty-only on the main dependency-sensitive slice, and the selected-versus-skipped harm pattern shows that the router spends budget on higher-consequence episodes. The new multi-budget diagnostic shows that consequence-aware rankings select higher exact branch harm than $p$-only routing at 10%, 25%, 50%, and 75% verification rates, with $p \times H$ strongest at tighter budgets and $H$-only strongest at larger budgets. The outcome effects remain small and not statistically conclusive under paired bootstrap intervals, so the result should not be read as a solved benchmark method. Its value is instead mechanistic: it isolates a specific failure of uncertainty-only allocation, namely that local hesitation is not the same object as downstream consequence.

The component ablation is the clearest evidence about mechanism. $H$-only is directionally strongest, $p \times H$ improves over $p$-only, and the uncalibrated product does not outperform the structural harm cue alone at the 50% budget point. Paired ablation intervals cross zero, so this is a directional mechanism result rather than statistical dominance. This is not a weakness of the allocation thesis. It is exactly the point that uncertainty-only routing misses: in this sandbox, the downstream-consequence signal carries more routing value than the public uncertainty proxy. At the same time, it limits the empirical claim about the particular multiplicative implementation. The theoretical value $q_t p_t H_t - c_t$ remains the natural expected loss-reduction target when components are well estimated, but multiplying two imperfect proxies need not improve finite-sample ranking. The held-out seed-split calibration diagnostic reinforces this distinction: calibration changes the selected set but does not improve aggregate outcomes on the evaluation split.

The learned PRM-style baselines clarify the boundary with process reward modeling. A learned scalar trained with counterfactual supervision can directionally improve over the transparent fixed score, although the intervals are not conclusive. This result strengthens the practical message rather than contradicting it. The framework identifies what should enter the allocation decision; a deployment may instantiate that decision with a transparent factorization, a learned composite model, or a hybrid, provided the routing objective accounts for consequence rather than local uncertainty alone.

The true `dep_sensitive_held_out` procedural diagnostic is deliberately not framed as a positive generalization result. In that small shifted generator, $p$-only is directionally strongest under the matched budget, and all intervals cross zero. This narrows the empirical claim: consequence-aware routing depends on harm cues that remain aligned with realized downstream consequence across generators.

Table 12: Adversarial relabel probe on the held-out seed split ($n$=200, exact budget 100). A fraction $\rho$ of highest-harm episodes has its harm score relabeled to the minimum evaluation harm score before routing.

| Condition | Attacked n | Attacked selected | Success | Irrev. fail | Avg. net reward |
|---|---|---|---|---|---|
| clean $p \times H$ | 0 | n/a | **0.795** | **0.205** | **3.172** |
| $\rho = 0.10$ | 20 | 0 | 0.740 | 0.260 | 1.273 |
| $\rho = 0.25$ | 50 | 0 | 0.655 | 0.345 | -1.276 |
| $\rho = 0.50$ | 100 | 17 | 0.665 | 0.335 | -1.064 |

Table 13: Correlated verifier-failure sensitivity on the held-out seed split ($n$=200, exact budget 100 by clean $p \times H$).

| Verifier model | Detected rate | Hard detect rate | Success | Irrev. fail | Avg. net reward |
|---|---|---|---|---|---|
| Uniform $q$=0.35 | 0.380 | **0.440** | **0.685** | **0.315** | **-0.707** |
| Correlated $q_{hard}$=0.10, $q_{easy}$=0.60 | 0.430 | 0.040 | 0.640 | 0.360 | -2.125 |
| $\Delta$ (correlated − uniform) | +0.050 | -0.400 | -0.045 | +0.045 | -1.418 |
| 95% CI | n/a | n/a | [-0.090, -0.005] | [0.005, 0.085] | [-2.737, -0.129] |

The stress tests are equally important. Harm-cue corruption and correlated verifier failures both degrade outcomes. These are not incidental implementation details. They show that consequence-aware verification requires reliable harm estimation and verifier reliability that does not collapse exactly on the highest-harm cases.

## 6 Threats to Validity and Limitations

Six limitations are most important. First, the experiments are controlled OpenClaw-facing sandbox evaluations rather than a full gateway-paired OpenClaw benchmark. The embedded trace in Appendix F validates interface compatibility, but it does not replace a broader end-to-end benchmark.

Second, the empirical harm signal is structural and hand-designed (Table 1). It should be understood as a proxy for $H_t$, not as a calibrated counterfactual estimator. The new alignment diagnostic shows strong pooled association because the cue separates coarse harm regimes, but only weak within-slice association on `dependency_sensitive`; this reinforces the interpretation of $\hat{H}_t$ as a coarse routing cue. The adversarial relabel probe shows that systematic corruption of this proxy can sharply degrade performance.

Third, the one-opportunity-per-episode design isolates routing cleanly but does not solve the fully online multi-step allocation problem in which early verification changes later proposals, future observations, and remaining budget.

Fourth, the strongest ablation is $H$-only. This supports the critique of uncertainty-only routing, but it also means the uncalibrated product $\hat{p}_t \hat{H}_t$ should not be treated as a universal empirical winner. The current evidence supports consequence-aware routing more strongly than it supports a specific uncalibrated multiplicative score.

Fifth, the verifier model remains simplified. The main sensitivity varies independent verifier detection probability, while the correlated-failure diagnostic shows that the same average verifier quality can be misleading if failures concentrate on high-harm episodes. This is why the deployment recipe recommends conditional $q_t$ estimation rather than relying on a single global verifier accuracy.

Sixth, the true `dep_sensitive_held_out` procedural variant is small and was added after the main `dependency_sensitive` results were locked. It is useful as a cross-generator stress test, but it should not be read as headline positive evidence. Under the exact top-$B$=50/100 matched budget, $p$-only is directionally strongest, while $H$-only and $p \times H$ remain close and all paired bootstrap intervals cross zero. Appendix D expands these construct, internal, external, statistical, adversarial-robustness, and reproducibility concerns.

# 7 Conclusion

When verification is scarce, the question is which decisions are worth checking. We argue that the answer depends on downstream consequence, not only local uncertainty. The likelihood that a proposed step is wrong and the harm of leaving that error unchecked are distinct quantities, and verification value depends on both. Controlled OpenClaw-facing sandbox experiments support this view in a deliberately bounded but informative way: harm-aware routing moves exact-budget outcomes in the predicted direction, multi-budget diagnostics show that consequence-aware rankings select higher-harm episodes across oversight budgets, the component ablation shows that downstream harm is the dominant signal in the dependency-sensitive sandbox, learned routers benefit from the same consequence information, and stress tests identify concrete failure modes when harm estimates or verifier reliability are corrupted. The most valuable check is therefore not necessarily the most uncertain step. It is the step whose error would be costly to leave in the trajectory, provided that the harm estimator and verifier are themselves reliable. This is the core contribution of the paper: budgeted verification should be evaluated as consequence-aware intervention allocation, not as uncertainty thresholding under another name.

## Broader Impact Statement

This paper proposes a decision framework for selectively invoking runtime verification of tool-using language-model agents. Because the framework governs *when* an agent's actions are screened before execution, its deployment interacts directly with the design of oversight for autonomous systems, and we identify several ethical considerations.

**Structural-metadata cues vs. chain-of-thought monitorability.** A verification policy that allocates based on a structural harm cue can in principle be defeated by an agent that learns to present high-harm actions in a form that scores low on the cue. This is related to but distinct from the chain-of-thought monitorability concern raised by Korbak et al. (2025): CoT monitorability relies on the agent's verbalised reasoning being readable, which is fragile because reasoning traces can be optimised against the monitor. Structural-metadata cues, by contrast, are tied to the action's declared effects on the environment rather than to the agent's verbalisation, and may therefore be *more* robust under optimisation pressure when the declared metadata is enforced by the tool API itself (e.g., a write-irreversible flag set by the tool, not by the agent) and *less* robust when the metadata is reported by the agent (e.g., a self-declared dependency count). The harm cue used in this paper combines both kinds of fields, and a deployment auditing the framework should distinguish them. Our adversarial probe (§4.11) directly tests the agent-reportable case. The framework as currently formulated assumes a non-adversarial or cooperative agent, and a deployment that does not satisfy this assumption risks *worse* oversight than uncertainty-only routing, because an adversarial agent need only suppress the harm cue rather than the broader uncertainty signal. The asymmetry has a sharper form than the prior literature notes: uncertainty-only routing reads a signal the adversary does not directly control (the model's own confidence head), while harm-aware routing reads a signal the adversary partly controls (declared metadata). A stronger adversarial evaluation, constructing an agent trained to minimise its own verification rate against a known harm estimator, remains future work beyond the relabel probe. Consequently, harm cues should ideally be tool-enforced or independently audited from execution logs, and the structural cue used here should not be the sole safeguard in adversarial or externally-facing deployments.

**$\beta$ as a governance decision.** The composite loss $\mathcal{L}_{\text{task}} + \beta \mathcal{L}_{\text{harm}}$ requires a choice of $\beta$ that is not technical: it reflects who bears the cost of task failures versus who bears the cost of irreversible side effects, and these are often different parties. Framing $\beta$ as an implementation detail, which we largely do in §3, risks obscuring that it is also a governance detail. In deployment contexts subject to regulatory or legal review, including medical decision support, financial transaction authorisation, legal document drafting, clinical coding, and automated code execution with external effects on production systems, a structural cue that has not been calibrated against realised harm outcomes should not be the sole input to a screening decision; we treat the framework as a way to *allocate* human oversight effort under a budget, not as a substitute for human oversight. If the verifier itself is another language model with correlated failure modes (Inan et al., 2025; Luo et al., 2025), the framework offers at most a marginal improvement, not a guarantee.

**Efficiency-enables-scale.** A framework that more efficiently allocates scarce human review effort can, in principle, lower the per-deployment cost of oversight enough to enable deployment of agent systems in settings that would not otherwise have passed oversight thresholds, shifting the question from "can we afford to review every action" to "can we afford to deploy this many autonomous agents." We do not think this framework makes that risk qualitatively worse than existing work on runtime safeguards (Xiang et al., 2025; Chen et al., 2025). More-efficient oversight is not automatically a safety improvement: its net effect depends on whether organisations reinvest the efficiency gain in deeper per-action review or use it to expand the set of actions that receive only cursory review. To balance these concerns, we identify one setting in which the framework is reasonably deployable today: *internal tool-use pipelines within a single organisation where agents are not subject to adversarial users, where $\beta$ can be set by a single accountable party, and where the verification budget is allocated across internal review tasks rather than externally-facing actions.* We do not recommend deployment in externally-facing or adversarial settings until the robustness concerns above have been substantively addressed. We have not identified positive societal impacts specific to this framework that are not already shared with existing work on selective prediction, conformal abstention (Ren et al., 2023), and process reward modelling (Choudhury, 2025); the positive case for the paper is scientific: a clearer decomposition of a decision problem that currently uses the wrong proxy, rather than a claim that the framework as implemented here is deployment-ready.

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

## A  Extended Discussion of the Myopic Decision Target

Remark 1 captures the allocation structure used by the online policy, but it deliberately avoids claiming optimality for the full sequential problem. Verifying step $t$ can change which actions are proposed later and therefore changes future $p$, $H$, $q$, and $c$ values. Spending budget at step $t$ also changes what is available later. A proper sequential treatment would require dynamic programming over a tractable state reduction or a regret analysis against a sequential oracle. We do not claim such a result. The contribution of the target $q_t p_t H_t - c_t$ is instead to identify the interpretable local factors that determine the expected value of checking a proposed step.

## B  Extended Method Discussion

The decomposition $\hat{q}_t \hat{p}_t \hat{H}_t - \lambda c_t$ estimates the components separately. In some settings, it may be preferable to learn a single composite score that approximates expected loss reduction directly. The PRM-style baselines in §4.9 are an example of this alternative. What should remain invariant is the decision structure: verification should be routed according to estimated intervention value, not according to local uncertainty alone. The current experiments also show why calibration matters. The uncalibrated product is transparent and useful, but it is not guaranteed to dominate either component when the uncertainty proxy is noisy.

## C  Detailed PRM Comparison

The PRM-style routers are trained on seeds 20–119 and evaluated on seeds 120–319 with an exact budget of 100 verifications. Both models are implemented as pure-Python logistic regressions with standardized tabular features because `sklearn` was not available in the environment. The features are $p$ score, harm score, $p \times H$ score, downstream dependency count, rollback difficulty, persistent-state risk, and downstream failure penalty. `PRM-task` predicts unverified failure. `PRM-value` predicts normalized positive reward gain from verification. The learned routers are directionally stronger than the transparent fixed product on this split, but paired bootstrap intervals cross zero. We therefore report them as learned diagnostic baselines, not as conclusive superiority claims.

## D  Full Threats to Validity Discussion

**Construct validity.** The theoretical $H_t$ is a counterfactual expectation over error and corrected continuations. The empirical harm cue is not that object. It is a structural proxy based on side-effect metadata, dependency count, and rollback availability. This is useful for testing whether downstream consequence should enter routing, but it does not solve general harm estimation.

**Internal validity.** The harm cue does not read the ground-truth outcome or realized future trajectory during routing. However, it was designed in the context of the sandbox, so the selected features may align with the implemented generator. The disjoint seed split checks calibration and learned baselines on held-out seeds. The additional `dep_sensitive_held_out` case is a separate small procedural variant with shifted harm cues, and its mixed result shows that cross-generator robustness depends on cue alignment rather than following automatically from the main sandbox result.

**External validity.** OpenClaw is useful as an execution substrate because it exposes tool state, workspace operations, and agent-facing commands. The current quantitative results nevertheless come from the controlled sandbox runner. A full gateway-paired OpenClaw benchmark and a port to another environment such as ToolEmu remain important next steps.

**Statistical validity.** The main exact-budget result uses $n{=}300$ paired seeds. Its paired bootstrap intervals for success, irreversible failure, and reward include zero, so the result is directional rather than statistically conclusive. The same caution applies to the component ablation, verifier-quality sensitivity, calibration diagnostic, PRM baselines, and true held-out procedural diagnostic. By contrast, the adversarial relabel

and correlated-verifier diagnostics show degradation with intervals excluding zero for the reported outcome differences.

**Adversarial robustness.** A structural harm cue is an attack surface. If an agent or upstream system can misreport high-harm actions as low-harm actions, consequence-aware routing can fail sharply, as Table 12 shows. In adversarial settings, the harm estimator should be audited and, where possible, grounded in tool-enforced metadata rather than agent-reported metadata.

**Verifier model.** The $q$-sensitivity table varies independent detection probability. The correlated-verifier diagnostic shows that a single average detection rate is insufficient. A verifier can have comparable average correction behavior while failing on the cases where correction matters most.

**Scope of allocation.** Each episode has one eligible verification point. This isolates the allocation question but does not address multi-step trajectories with interacting verification opportunities. Extending the method to fully online allocation is a separate sequential decision problem.

**Reproducibility.** The anonymized supplementary repository records the artifacts needed to audit the reported tables: runlists, raw CSV outputs, summary files, bootstrap summaries, locked artifact folders, scripts for each completed diagnostic, checksum records, and an OpenClaw interface folder with reset/hint and finalize scripts, prompts, templates, and saved embedded traces. This directly addresses reproducibility at the scope of the revised claims. The paper does not claim a full gateway-paired OpenClaw benchmark over unconstrained natural tasks; it claims a controlled OpenClaw-facing agent evaluation designed for paired, matched-budget allocation analysis.

## E   Bootstrap and Diagnostic Details

Table 14 summarizes which results are statistically conclusive under the locked bootstrap analyses. This table is included to prevent overreading small directional differences.

Table 14: Summary of bootstrap interpretation across completed diagnostics.

| Diagnostic | Bootstrap status | Manuscript interpretation |
|---|---|---|
| Main exact-budget Table 2 | Outcome intervals cross zero | Directional evidence only |
| Multi-budget Table 3 | Not a significance table | Budget-dependent allocation diagnostic |
| H-proxy alignment Table 5 | Correlation diagnostic | Coarse proxy alignment, not calibrated $H_t$ |
| Component ablation Tables 6–7 | Pairwise intervals cross zero | $H$ is directionally dominant, not statistically dominant |
| Verifier-quality Table 8 | Intervals cross zero | Sensitivity diagnostic |
| Joint calibration Table 9 | Intervals cross zero | Calibration changes selection but not aggregate outcomes |
| PRM baselines Table 10 | Intervals cross zero | Learned routers directionally improve |
| True held-out procedural Table 11 | Intervals cross zero | Small stress test, not cross-generator dominance |
| Adversarial relabel Table 12 | Degradation intervals exclude zero | Supported failure mode |
| Correlated verifier Table 13 | Degradation intervals exclude zero | Supported failure mode |

## F    OpenClaw Embedded-Agent Trace Validation

As an interface validation rather than a quantitative result, we ran a representative embedded-agent trace through `openclaw agent --agent main -m`. The trace successfully accessed the workspace, executed shell commands, and completed a dependency-sensitive harm-aware episode with `verify_used`=1, `success`=1, `irreversible_failure`=0, exact branch harm 34.0, and total reward 9.2. This trace evidences interface compatibility. The quantitative claims in the main text rest on the controlled OpenClaw-facing sandbox tables.

## G    Supplementary Toy Environment

The toy environment is a small synthetic sandbox designed to complement the OpenClaw-facing experiments by isolating the mechanism in a transparently inspectable setting. Each episode consists of a short sequence of discrete decisions, some of which write to a shared state that subsequent decisions read from. Errors on state-writing steps can be recoverable, if later steps overwrite them, or irreversible, if later steps commit to the incorrect value. A tunable uncertainty signal $u_t$ controls signal-consequence discordance. The toy environment is useful for cheap stress tests of the ranking principle, but it does not substitute for the OpenClaw-facing experiments because it lacks a real agent interface and realistic tool-state propagation.

## H    Extended Future Work

The most direct empirical extension is a full gateway-paired OpenClaw benchmark, followed by a port to another agent safety environment such as ToolEmu. The structural harm cue should be replaced or complemented by learned estimators trained from counterfactual rollouts, trajectory-level harm labels, or human review. The current verifier diagnostics also suggest that $q_t$ should be modeled as an episode-dependent reliability term, especially when the verifier and base agent share the same model family. Finally, the one-opportunity-per-episode setting should be generalized to trajectories with multiple verification opportunities, where spending budget early changes later observations, proposals, and remaining budget.

