# OpenReview forum: "Verify What Matters: Budgeted Verification for Tool-Using Agents under Counterfactual Downstream Harm"
_TMLR — Under review for TMLR_

### Review · Reviewer_P6zt · 2026-05-03

**Summary Of Contributions:**

This paper studies how to verify tool-using agent behavior when verification is costly. Specifically, the paper formulates budgeted-verification for tool-using agents as an intervention-allocation problem. Traditional approaches to this problem often consider confidence/uncertainty only. In this work, the authors propose a theoretical framework that factos verifier efficacy, local error probability, counterfactual downstream harm, and intervention cost into step-level value function. Empirically, the paper reports a four-episode mechanism pilot on an Open-Claw-based sandbox.

**Audience:**

No

**Audience Explanation:**

See above. While I think many TMLR's audience would be interested in the topic, the current paper is below the bar that produces convincing results and arouse TMLR's audience's interest in reading the whole paper.

**Broader Impact Concerns:**

I do not see ethical concerns of this work.

**Claims And Evidence:**

No

**Claims Explanation:**

The paper studies the tool-using agent verification problem in a highly practical setup . While this is a very important problem and the formulation of budgeted verification provides a clean construct, I think the paper has major limitations that need to be addressed.

1. The paper spends a large portion of the article to introduce the notations and maths of the framework, which is based on probablity and expectation. However, the major challenge of how to estimate $q_t$ $p_t$ $H_t$ in praction remains unaddressed. The paper mentions "a practioner deploying this framework in a new setting should caliberate the three component estimators jointlly on held-out data rather than in isolation", it is unclear how the practioner should actually do it. From this perspective, I am unsure what value this framework offers beyond laying out the factors including verifier efficacy, local error probability, counterfactual downstream harm, and intervention cost.
2. The emperical experiment part is very insufficient. The paper conducts experiments using an OpenClaw-based sandbox but only includes four episodes. While the paper discusses a lot that this experiment is only to support the theoretical framework so it requires less data points than common benchmarks. That being said, I think four data points are too limited especially given that Section 4 draws a lot of conclusions based on them. Also, suppose the sandbox is set up, I think it won't be too challenging to enlarge the sample size to actually construct a benchmark for the field to facilitate the study of this research problem.
3. The paper is not very well written. There are many small paragraphs in the framework description part, which breaks the flow and reads like AI-generated. The description of experiment detail is insufficient; and the tables and figures can be futher improved.

**Requested Changes:**

1. Besides laying out the framework formulation, the paper shall discuss how to estimate each component in practice and what metrics to optimize for on the held-out set.
2. Enhance the empericial experiments by enlarging the sample size or constructing a benchmark to help the community study this problem.
3. Improve the paper writing by reducing the number of small paragraphs and including more implementation details of the Open-Claw sandbox and how the emperical experiment is conducted.

---

> ### Author Response · Authors · 2026-06-30
> **Revision Summary and Author Response**
>
> We thank the reviewer for the direct and useful feedback. We agree that the original submission placed too much weight on the framework notation while not giving enough practical guidance on how the quantities should be estimated, calibrated, and evaluated. We also agree that a four-episode pilot was insufficient. The revision addresses these concerns by expanding the empirical evaluation, adding operational-estimation details, and rewriting the exposition for a smoother and more implementation-oriented presentation.
> How to estimate the framework components in practice. We added a detailed operational-estimation section and a deployment recipe. The revised manuscript specifies how p_t, H_t, q_t, and c_t are instantiated in the sandbox and also describes practical alternatives: public confidence, log-probability margins, self-consistency, step-error classifiers, structural harm cues, learned harm estimators from counterfactual rollouts, and conditional verifier-efficacy estimates.
> Calibration and validation objective. We added a six-step recipe covering harm definition, β selection, estimator selection, joint calibration on held-out data, fixed-budget selection, and threshold tuning on the validation objective L_task + β L_harm. The recipe explains why joint calibration matters for a multiplicative score and why the budget must be fixed before threshold tuning.
> Expanded empirical evaluation. We expanded the empirical evaluation beyond the original four episodes. The revised paper now reports n=300 controlled sandbox comparisons for the main settings, with exact-budget evaluation, cross-slice behavior, multi-budget diagnostics, component ablations, verifier-quality sensitivity, held-out seed-split calibration, learned PRM-style baselines, a small true held-out procedural diagnostic, and two stress tests. This is still not claimed to be a full benchmark, but it provides substantially more evidence than the original pilot.
> Implementation details of the OpenClaw-facing sandbox. We rewrote the experimental setup to describe the controlled OpenClaw-facing sandbox, one eligible verification point per episode, the verifier intervention mechanism, the reward/cost structure, the slices, the paired-seed protocol, exact-budget comparisons, shared-threshold diagnostics, and bootstrap intervals. The supplementary repository includes the sandbox runner and scripts needed to reproduce the reported tables.
> Writing and flow. We revised the manuscript structure to improve readability and reduce the impression that the framework is only notation. The paper now includes a clearer scope-of-evidence paragraph, a more concrete deployment recipe, a rewritten experimental setup, a central ablation discussion, and a limitations section that explicitly separates the decision-theoretic target from the structural proxy used in the experiments.
> Tables and figures. We improved the empirical presentation by adding exact-budget tables, a multi-budget table, component-ablation confidence intervals, verifier-quality sensitivity, learned-router baselines, and stress-test tables. We also clarified that boldface indicates directional preference within the relevant matched comparison block and is not a significance marker.
> We hope the revised manuscript makes the value of the framework clearer: the contribution is not only the decomposition itself, but a concrete way to evaluate scarce verification as consequence-aware intervention allocation under fixed budgets. We also narrowed the claims so that the empirical evidence is presented as a controlled diagnostic study rather than as a solved benchmark method.

---

### Review · Reviewer_Qd1P · 2026-05-24

**Summary Of Contributions:**

This paper investigates the problem of budgeted verification for tool-using agents, framing the verification as an intervention-allocation problem. The core claim is that the value of verification factors as $V_t = q_t p_t H_t - c_t$, where $H_t$ is counterfactual downstream harm. A hand-engineered structural harm cue is compared against 3 baselines on a 4-episode pilot in an OpenClaw-based sandbox, showing it intercepts irreversible task failures that uncertainty-only methods would miss, although the pilot contrast is consistent with noise.

## Strengths:
1. The paper studies a timely and relevant question, and addresses a bottleneck in the real-world deployment of agents in environments with irreversible states.
2. The separation of $p_t$ and $H_t$ as distinct decision inputs is sound, which identifies when uncertainty-only routing fails and reduces uncertainty-only routing to a special case.
3. The 3 planned follow-up comparisons (matched-budget, cross-slice robustness, PRM baseline) are specified with policy definitions, sample sizes, test statistics and decision implications.

## Weaknesses:
1. The pilot-scale results do not distinguish the proposed mechanism from noise by the paper's own test, which provides no reliable evidence.
2. The authors use hand-crafted episodes rather than samples from a natural task distribution for the predicted configuration.
3. Table 1 uses values inconsistent with the reward function stated in the supplementary, and the authors leave reconciliation to camera-ready, which makes it unverifiable for review.
4. While it is noted that numerical placeholders in the supplementary material are marked for camera-ready completion, the missing content is more than numerical constants and undermines necessary verification.
5. The comparison between the two selective policies is confounded by the verify rate (3/4 and 1/4). A matched-budget control is required.

**Audience:**

Yes

**Audience Explanation:**

The paper studies a timely and relevant question, and addresses a bottleneck in the real-world deployment of agents in environments with irreversible states. The decoupling of local error probability from downstream harm magnitude is a sound conceptual contribution, which other researchers in the related field could find interesting.

**Broader Impact Concerns:**

None. The paper already has a Broader Impact Statement section that adequately addresses several concerns in detail.

**Claims And Evidence:**

No

**Claims Explanation:**

While the proposed conceptual claims in the paper are generally logically sound, the empirical evidence is insufficient.

1. The pilot results cannot distinguish the proposed mechanism from noise by the paper's own test. The key contrast is 0/4 vs 2/4 irreversible failures, yielding Fisher exact $p≈0.43$, which is consistent with the null hypothesis that the two policies perform identically at $n=4$. This provides no reliable mechanism evidence as stated.

2. 4 hand-crafted episodes rather than samples from a natural task distribution are used and are constructed to instantiate the predicted configuration, so the outcome of Table 1 merely follows from the episode design, which makes them invalid as independent empirical evidence for the framework.

3. There is discrepancy between Table 1's numeric values and the stated reward function in the supplementary and reconciliation is deferred to camera-ready. This makes the empirical results unverifiable and unable to support the claim that "robust to this reconciliation.

4. Placeholders in the supplementary material exceed what the authors acknowledge and prevent independent verification.

5. The comparison between the two selective policies is confounded by the verify rate (3/4 and 1/4). A matched-budget control is required for evaluating the observed performance difference.

6. The OpenClaw sandbox is not released, and there is no result reported for the toy environment, which makes the work lack reproducibility.

**Requested Changes:**

## Critical:
1. Reconcile the Table 1 values to match the stated reward function and the results.

2. Complete the supplementary material, filling in the necessary placeholders, such as task descriptions, correction actions, actual numeric values, etc.

3. Either expand the experimental evaluation beyond the 4-episode hand-crafted pilot, or reframe the empirical claims for accuracy.

4. Conduct a matched-budget comparison at an equal verify rate as the current comparison in the paper is confounded and could reflect verification volume rather than routing quality.

## Recommended:
1. Either release the OpenClaw sandbox or fully report the toy environment experiments for reproducibility.

2. Specify and describe the held-out set used for the harm-cue threshold selection and the feature weights.

---

> ### Author Response · Authors · 2026-06-30
> **Revision Summary and Author Response**
>
> We thank the reviewer for the detailed critique. The review correctly identified several weaknesses in the original submission: the pilot was too small to support empirical claims, the selective-policy comparison confounded routing quality with verification volume, the supplementary material was incomplete, and the reward/table reconciliation was not sufficiently auditable. We have revised the paper and supplementary material to address these issues directly.
> Expansion beyond the four-episode pilot. We replaced the original empirical emphasis with a controlled OpenClaw-facing sandbox evaluation at n=300 for the main sandbox comparisons. The four-episode pilot is no longer the empirical basis for the paper’s claims. The revised experiments include exact-budget dependency-sensitive evaluation, cross-slice diagnostics, multi-budget analysis, component ablations, verifier-quality sensitivity, calibration/held-out diagnostics, learned PRM-style baselines, and stress tests.
> Matched-budget comparison. We fixed the verification-rate confound directly. In the primary dependency-sensitive comparison, uncertainty-only routing and harm-aware routing both verify exactly 150 of 300 episodes. The manuscript now separates exact-budget comparisons, where verification volume is fixed, from shared-threshold cross-slice diagnostics, where the goal is to study routing behavior across regimes.
> Reward consistency and table reconciliation. We revised the empirical section so that the reported tables are generated from the controlled sandbox runner and locked per-episode outputs. The supplement now contains raw per-episode files, aggregation scripts, bootstrap scripts, locked summary files, and checksum records, allowing the reported values to be audited from the released artifacts. We removed the previous reliance on unresolved camera-ready reconciliation.
> Completion of supplementary material. We completed the supplementary package and removed placeholder-style material. The anonymized repository now includes the controlled OpenClaw-facing sandbox runner, policy runlists, raw outputs, aggregation and bootstrap scripts, calibration and PRM scripts, example configurations, OpenClaw-facing interface artifacts, reset/hint and finalize scripts, episode templates, prompts, runlists, saved embedded trace outputs, and checksum records.
> Reproducibility and OpenClaw-facing artifacts. The revised manuscript includes a reproducibility paragraph in the experimental setup and an appendix note mapping the quantitative claims to the supplementary artifacts. We also added an OpenClaw embedded-agent trace validation as interface evidence, while making clear that the quantitative claims are tied to the controlled OpenClaw-facing sandbox rather than a full gateway-paired OpenClaw benchmark.
> Held-out threshold selection and feature weights. We added an operational-estimation table specifying the public uncertainty signal, structural harm cue, verifier efficacy, verification cost, and exact branch harm used for analysis only. The structural harm cue and its feature weights are now stated explicitly. We also describe the disjoint seed split used for calibration and learned routing diagnostics: seeds 20–119 for calibration/training and seeds 120–319 for evaluation.
> Claim narrowing. We agree that the original empirical claims needed to be reframed. The revised manuscript states that the main outcome intervals overlap zero and presents the evidence as directional or diagnostic. The supported conclusion is now that uncertainty-only routing is incomplete when downstream consequences vary, not that the current p_t × H_t implementation establishes quantitative superiority in general.
> We appreciate the reviewer’s insistence on matched-budget evaluation and reproducible evidence. These points substantially improved the revision, and the revised paper now treats allocation quality separately from verification volume and ties the reported tables to auditable supplementary artifacts.

---

### Review · Reviewer_aBPb · 2026-06-24

**Summary Of Contributions:**

This paper studies budgeted verification for tool-using agents. The main idea is that verification should not be allocated only by local uncertainty, because two similarly uncertain tool actions can have very different downstream consequences. The paper formalizes a myopic verification value. Empirically, the authors evaluate a controlled OpenClaw-facing sandbox with one eligible verification point per episode. The results show that consequence-aware routing selects higher-harm episodes than uncertainty-only routing under the same verification budget, although most outcome differences are small and statistically inconclusive.

Strengths
- The problem is well motivated. Costly verification is a central issue for tool-using agents, and the paper correctly distinguishes local uncertainty from downstream consequence.
- The conceptual decomposition is clear and useful. It helps explain when uncertainty-only routing is a reasonable approximation and when it is likely to fail.
- The experiments are careful about matched verification budgets and avoid overclaiming large performance gains from small differences.
- The paper includes useful diagnostics, including component ablations, held-out seed splits, learned PRM-style routers, harm-cue corruption, and correlated verifier-failure stress tests.

Weaknesses
- The empirical setting is still restricted: one verification opportunity per episode, a controlled sandbox rather than a full OpenClaw benchmark, and small n for the main and held-out comparisons.
- The operational harm signal is a hand-designed structural proxy, not an estimator of the counterfactual H_t defined in the theory. This gap limits how much the experiments validate the formal objective.
- The main positive outcome differences are directional rather than statistically conclusive, and the H-only ablation is stronger than the proposed p x H score in the main dependency-sensitive slice.
- The held-out procedural diagnostic is mixed, with p-only directionally strongest, so the robustness of consequence-aware routing under generator shift remains unclear.

**Additional Comments:**

The authors should add a short "Discussion / Connection to Prior Work" subsection comparing this work with the most relevant recent agentic reasoning and process-reward papers. This does not require new experiments for all of them, but it would clarify the paper's position.

- AlphaApollo: A System for Deep Agentic Reasoning (arXiv 2026) is relevant because it also studies tool-assisted, multi-turn agentic reasoning with verification in the loop. The most useful comparison would be conceptual: AlphaApollo uses tool-assisted verification to improve iterative reasoning, while this paper asks which tool actions should receive verification under a fixed budget. The authors could discuss whether their p/H/q decomposition could allocate AlphaApollo-style verification calls across propose-judge-update rounds.
- From Passive to Active Reasoning: Can Large Language Models Ask the Right Questions under Incomplete Information? (ICML 2025) is relevant because it evaluates interactive agents that must gather information and notes that verifier quality limits search-based strategies. The authors should contrast AR-Bench's active information acquisition setting with their intervention-allocation setting and suggest evaluating budgeted verification on AR-Bench-like tasks where questions or tool calls have unequal downstream value.
- The Easy, the Hard, and the Learnable: Confidence and Difficulty-Adaptive Policy Optimization for LLM Reasoning (ICML 2026) is somewhat related through fixed-budget allocation: CoDaPO allocates training effort by confidence and difficulty, whereas this paper allocates runtime verification by error probability and downstream harm. A short comparison would help clarify that both works are budget-allocation methods but over different resources and objectives.
- Landscape of Thoughts: Visualizing the Reasoning Process of Large Language Models (ICLR 2026) is less central but relevant to verifier construction, since it adapts trajectory features into a lightweight verifier. The authors could mention it as an example of learned trajectory-level verification signals, distinct from structural harm estimation.

**Audience:**

Yes

**Audience Explanation:**

The paper should interest researchers working on LLM agents, process supervision, selective prediction, runtime safety, and cost-aware evaluation. The central message is simple but important: in state-changing agent trajectories, the most uncertain action is not necessarily the action most worth verifying. Even with bounded empirical evidence, the formulation and diagnostics are useful for thinking about oversight allocation.

**Broader Impact Concerns:**

The broader impact discussion is appropriate.

The main concern is that a harm-aware router can become an attack surface if the harm metadata is agent-reported or otherwise manipulable. The paper should emphasize that consequence-aware verification is safer when harm cues are grounded in tool-enforced metadata or independently audited logs and should not be deployed as the sole safeguard in adversarial or externally facing settings.

**Claims And Evidence:**

Yes

**Claims Explanation:**

The evidence supports the paper's narrower claim that uncertainty-only routing is incomplete when downstream consequences vary. The matched-budget design is appropriate, and the selected-versus-skipped harm diagnostics show that the harm-aware router spends budget on higher-consequence episodes. The authors are also transparent that the main outcome intervals cross zero and that the current p x H implementation is a diagnostic instantiation rather than a universally dominant router.

However, the evidence is not strong enough for broad claims about budgeted verification in realistic tool-using agents. The experiments use a one-opportunity sandbox, and the harm cue is manually engineered from metadata rather than learned or estimated from counterfactual rollouts. The paper should keep its claims at the level of a useful framework and controlled diagnostic study unless it adds a full multi-step benchmark or stronger external validation.

**Requested Changes:**

- Clarify the scope of the main claim throughout the abstract, introduction, and conclusion. The current evidence supports "downstream consequence is a useful and sometimes necessary routing signal in this controlled sandbox," not a general claim that p x H routing reliably improves tool-agent verification.
- Strengthen the connection between theoretical H_t and the structural harm proxy. At minimum, report the correlation or rank correlation between the proxy H and exact branch harm across slices, and separately for selected and skipped episodes. A stronger version would compare structural H against a learned harm estimator trained on held-out counterfactual branch-harm labels.
- Reframe the component ablation more centrally. Since H-only outperforms p x H in Table 4 and p-only is strongest in the small held-out procedural diagnostic, the paper should explicitly state that the current empirical win is for consequence-aware routing, not for the multiplicative product as implemented.
- Add a multi-budget curve rather than only the 50% exact-budget point. For example, verify rates in {10%, 25%, 50%, 75%}, reporting success, irreversible failure, net reward, and selected branch harm.
- Report paired tests or confidence intervals for the H-only vs p-only and H-only vs p x H ablations, not only for the main uncertainty-only comparison.
- Clarify how beta would be chosen in practice and whether the same routing conclusions hold for multiple beta values.
- Add a more explicit discussion of verifier reliability estimation. Table 10 shows that average q can be misleading, so a deployment recipe should include how to estimate q conditional on action type, harm class, or state.
- Add a small external-validity experiment if feasible. A useful target would be an OpenClaw or ToolEmu-style multi-step setting with multiple candidate verification points per trajectory, comparing uncertainty-only, H-only, p x H, and a learned PRM-value router under the same verification-count budget.

---

> ### Author Response · Authors · 2026-06-30
> **Revision Summary and Author Response**
>
> We thank the reviewer for the thoughtful and detailed review, and especially for the constructive guidance on how to scope the contribution. We agree with the reviewer’s central assessment that the evidence supports a bounded claim about controlled consequence-aware allocation, rather than a broad claim that the current p_t × H_t score reliably improves verification in all realistic tool-agent settings. The revision was organized around that distinction.
>
> Scope of the main claim. We revised the abstract, introduction, scope-of-evidence paragraph, discussion, limitations, and conclusion to make the claim deliberately bounded. The paper now states that the experiments are controlled OpenClaw-facing sandbox evaluations, not a full gateway-paired OpenClaw benchmark, and that the evidence supports consequence-aware allocation as a diagnostic principle rather than universal dominance of the transparent product score.
>
> Connection between theoretical H_t and the operational structural proxy. We added an explicit discussion distinguishing theoretical counterfactual downstream harm H_t from the structural harm cue used in the sandbox. We also added a structural-proxy alignment diagnostic reporting Pearson and Spearman correlations between the proxy and exact branch harm across slices, together with selected-versus-skipped branch-harm summaries for the harm-aware router. The manuscript now treats the structural cue as a coarse correlate of H_t, not as a calibrated counterfactual estimator.
>
> Component ablation and the role of H-only. We reframed the component ablation centrally. The revised manuscript explicitly states that H-only is directionally strongest at the 50% dependency-sensitive budget point, that p_t × H_t improves over p-only but does not outperform H-only there, and that the empirical support is stronger for consequence-aware routing than for the particular uncalibrated multiplicative implementation.
>
> Multi-budget curve. We added a multi-budget diagnostic at 10%, 25%, 50%, and 75% verification rates on the dependency-sensitive slice. The table reports success, irreversible failure, average net reward, and selected branch harm for p-only, H-only, and p_t × H_t rankings under exact top-B budgets.
>
> Paired confidence intervals for ablations. We added paired bootstrap confidence intervals for H-only versus p-only, H-only versus p_t × H_t, and p_t × H_t versus p-only at the exact B=150/300 budget. These intervals cross zero, and the manuscript states that the ablation evidence is directional rather than statistically conclusive.
>
> Choice of β and deployment guidance. We added a paragraph on choosing the harm weight β as a stakeholder/governance choice and included a concrete cost-ratio example. We also added a deployment recipe specifying harm definition, β selection, estimator selection, joint calibration, fixed-budget threshold tuning, and the validation objective L_task + β L_harm.
>
> Verifier reliability estimation. We added an explicit discussion that a single average q_t can be misleading when verifier failures concentrate on high-harm actions. The deployment recipe now recommends estimating or auditing q_t conditionally on action type, side-effect class, rollback availability, harm class, state, and verifier/model family. The correlated-verifier failure stress test is used to demonstrate this failure mode.
>
> External-validity diagnostic. We added a small true held-out procedural diagnostic with shifted structural-harm cues and report the result as mixed: p-only is directionally strongest and all intervals cross zero. We use this result to narrow, not inflate, the claim about cross-generator robustness, and we state full multi-step benchmark evaluation as future work.
>
> Broader impact and attack surface. We expanded the broader impact discussion to emphasize that harm-aware routing can become an attack surface when metadata is agent-reported or manipulable. The revised manuscript recommends grounding harm cues in tool-enforced metadata or independently audited logs, and it states that the structural cue should not be deployed as the sole safeguard in adversarial or externally facing settings.
>
> Connection to recent agentic reasoning and process-reward work. We added a related-work subsection comparing the paper to recent agentic reasoning and allocation systems, including AlphaApollo, AR-Bench, CoDaPO, and Landscape of Thoughts. The revised text clarifies that these works study complementary problems, while our paper focuses on allocating runtime verification under a fixed oversight budget.
>
> We appreciate the reviewer’s suggestion to keep the claims at the level supported by the evidence. The revised paper follows that recommendation: it presents a controlled diagnostic study and a reproducible framework for consequence-aware allocation, while explicitly leaving full multi-step benchmark evaluation and stronger external validation to future work.